# Corner Flows Induced by Surfactant-Producing Bacteria *Bacillus subtilis* and *Pseudomonas fluorescens*

Yuan Li,[a,b] Joseph E. Sanfilippo,[c] ⓘ Daniel Kearns,[d] ⓘ Judy Q. Yang[a,b]

[a]Saint Anthony Falls Laboratory, University of Minnesota, Minneapolis, Minnesota, USA
[b]Department of Civil, Environmental, and Geo-Engineering, University of Minnesota, Minneapolis, Minnesota, USA
[c]Department of Biochemistry, University of Illinois at Urbana—Champaign, Urbana, Illinois, USA
[d]Department of Biology, Indiana University, Bloomington, Indiana, USA

**ABSTRACT**    A mechanistic understanding of bacterial spreading in soil, which has both air and water in angular pore spaces, is critical to control pathogenic contamination of soil and to design bioremediation projects. A recent study (J. Q. Yang, J. E. Sanfilippo, N. Abbasi, Z. Gitai, et al., Proc Natl Acad Sci U S A 118:e2111060118, 2021, https://doi.org/10.1073/pnas.2111060118) shows that *Pseudomonas aeruginosa* can self-generate flows along sharp corners by producing rhamnolipids, a type of biosurfactants that change the hydrophobicity of solid surfaces. We hypothesize that other types of biosurfactants and biosurfactant-producing bacteria can also generate corner flows. Here, we first demonstrate that rhamnolipids and surfactin, biosurfactants with different chemical structures, can generate corner flows. We identify the critical concentrations of these two biosurfactants to generate corner flow. Second, we demonstrate that two common soil bacteria, *Pseudomonas fluorescens* and *Bacillus subtilis* (which produce rhamnolipids and surfactin, respectively), can generate corner flows along sharp corners at the speed of several millimeters per hour. We further show that a surfactin-deficient mutant of *B. subtilis* cannot generate corner flow. Third, we show that, similar to the finding for *P. aeruginosa*, the critical corner angle for *P. fluorescens* and *B. subtilis* to generate corner flows can be predicted from classic corner flow theories. Finally, we show that the height of corner flows is limited by the roundness of corners. Our results suggest that biosurfactant-induced corner flows are prevalent in soil and should be considered in the modeling and prediction of bacterial spreading in soil. The critical biosurfactant concentrations we identify and the mathematical models we propose will provide a theoretical foundation for future predictions of bacterial spreading in soil.

**IMPORTANCE**    The spread of bacteria in soil is critical in soil biogeochemical cycles, soil and groundwater contamination, and the efficiency of soil-based bioremediation projects. However, the mechanistic understanding of bacterial spreading in soil remains incomplete due to a lack of direct observations. Here, we simulate confined spaces of hydrocarbon-covered soil using a transparent material with similar hydrophobicity and visualize the spread of two common soil bacteria, *Pseudomonas fluorescens* and *Bacillus subtilis*. We show that both bacteria can generate corner flows at the velocity of several millimeters per hour by producing biosurfactants, soap-like chemicals. We provide quantitative equations to predict the critical corner angle for bacterial corner flow and the maximum distance of the corner spreading. We anticipate that bacterial corner flow is prevalent because biosurfactant-producing bacteria and angular pores are common in soil. Our results will help improve predictions of bacterial spreading in soil and facilitate the design of soil-related bioremediation projects.

**KEYWORDS**    *Bacillus subtilis*, bacterial spreading, *Pseudomonas fluorescens*, biosurfactant, contact angle, corners, hydrocarbons, hydrophobicity, rhamnolipids, soil, soil microbiology, surfactin

Address correspondence to Judy Q. Yang, judyyang@umn.edu.

The authors declare no conflict of interest.

The movement of bacterial cells in soil regulates soil biogeochemical cycles and plays a critical role in soil carbon decomposition (1, 2). Such movement also controls the transport of pathogenic bacteria from fecal waste to drinking water reservoirs, posing risks to human health (3). In addition, the spread of bacterial cells in soil controls the efficiency of many soil remediation projects in which contaminant-degrading bacteria, such as *Bacillus* spp., are injected into soil (4–6). A mechanistic understanding of how bacterial cells spread in soil is needed to predict soil biogeochemical cycles and improve soil quality, and yet, such understanding is currently incomplete.

In most studies, bacterial spreading was attributed to advection and hydrodynamic dispersion (7, 8), filtration (9, 10), adsorption (11, 12), and desorption (11), as well as bacterial motility (12). Many empirical studies also show that the addition of surfactants enhances bacterial spreading in porous media (13, 14). In addition to the above-mentioned bacterial transportation mechanisms, a recent study (15) discovered a new bacterial spreading mechanism by showing that *Pseudomonas aeruginosa*, a major human pathogen and bacterium found in soil, can self-generate flows along sharp corners and spread in a synthetic hydrophobic soil by producing biosurfactants that change the solid surface into a hydrophilic one.

Corner flow refers to the spread of liquid along the corner where two solid surfaces meet. Corner flow generated by pure wetting liquid is a classical fluid mechanics problem (16, 17). It is induced by the bending of the air-water interface along the corner, which produces a force along the corner that drives a flow (16–19). Such flow only occurs for hydrophilic surfaces and corners with angles that are less than a critical value (20, 21). In a recent study, Yang et al. (15) showed that *P. aeruginosa* can generate corner flows on an initially hydrophobic surface (i.e., for initially nonwetting liquid) because the cells generate rhamnolipids, which turn hydrophobic into hydrophilic surfaces (22, 23). Hydrophobic soil is ubiquitous and is caused by the coating of organic matter (including vegetation exudates, wildfire products, and oil residues) on the surface of sediment (24–26). Here, we use polydimethylsiloxane (PDMS), transparent organic polymers covered with nonaromatic hydrocarbon functional groups, to represent hydrocarbon-covered hydrophobic soil, which is common near oil fields (27–29).

The goal of this study was to test our hypothesis that, in addition to rhamnolipids and *P. aeruginosa*, other types of biosurfactants and biosurfactant-producing bacteria could also generate corner flows in hydrophobic soil. To achieve this goal, we investigated corner flows generated by the two most common types of biosurfactants, rhamnolipids and surfactin, and two common soil bacteria, *Pseudomonas fluorescens* (30, 31) and *Bacillus subtilis* (32, 33). Rhamnolipids, a classic type of biosurfactants, are glycolipids with carbohydrates covalently linked to lipids (Fig. 1A, bottom right). Surfactin, another common type of biosurfactant, consists of a peptide loop (Fig. 1B, bottom right). *P. fluorescens* and *B. subtilis* are bacteria that produce rhamnolipids (34, 35) and surfactin (36, 37), respectively. Both bacteria are plant growth-promoting rhizobacteria (38) and ubiquitous in soil (39, 40). *B. subtilis* is a Gram-positive soil bacterium (41), while *P. fluorescens* is a Gram-negative soil bacterium (42). The distinct structures of both surfactants and soil bacteria enable us to test our hypothesis and extend the previous study (15) to the soil environment.

In this study, we first show that both types of biosurfactants, purified rhamnolipids and surfactin, can generate corner flows by making the initial hydrophobic surface hydrophilic. We quantify the critical concentrations of rhamnolipids and surfactin required to generate corner flows. Second, we show that rhamnolipid-producing *P. fluorescens* and surfactin-producing *B. subtilis* bacteria can generate corner flows, while a surfactin-deficient mutant of *B. subtilis* cannot. We further show that the contact angle of the solution dictates the critical corner angle for the corner flow. In addition, we show that the maximum height of corner flows generated by bacteria is controlled by the roundness or cutoff of the corner tip and can be predicted by the capillary theory. Finally, we discuss how our results will help improve predictions of bacterial spreading in soil and facilitate the design of soil-related bioremediation projects.

## RESULTS

**Biosurfactant types, concentrations, and corner flows.** To test the conservation of surfactant-driven corner flow, we examined whether the biosurfactants, rhamnolipids and

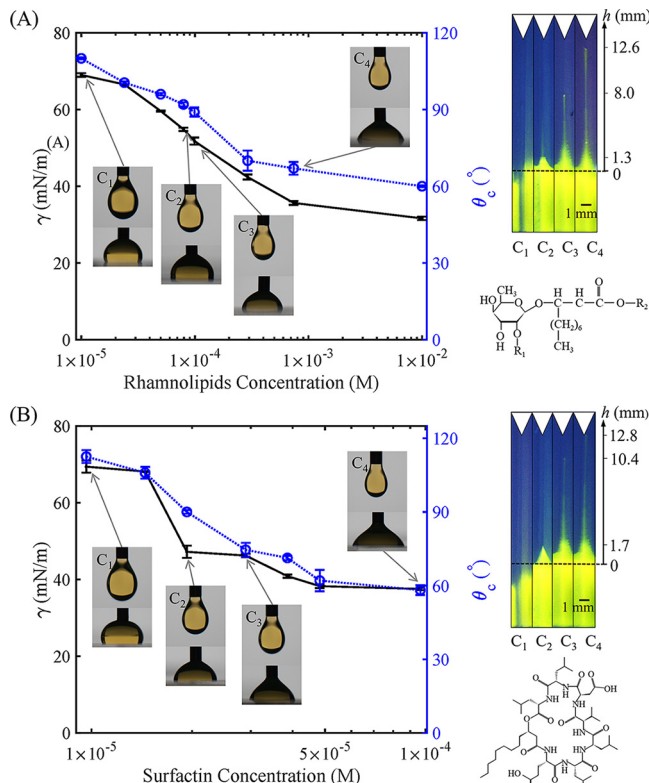

**FIG 1** The surface tension $\gamma$ (black, left $y$ axis) and contact angle $\theta_c$ (blue, right $y$ axis) values for rhamnolipids solution (A) and surfactin solution (B) at different concentrations. The insets are representative images of pendant droplets to measure surface tension and of moving droplets on a PDMS surface to measure contact angle (see Materials and Methods for details). The top right images show the corner flows generated by rhamnolipids and surfactin at representative concentrations ($C_1$, $C_2$, $C_3$, and $C_4$) indicated in the graphs. The bottom right images show representative chemical structures of rhamnolipids and surfactin. The error bars represent the standard errors of measurements of three droplets at each concentration.

surfactin, could generate corner flows on hydrophobic surfaces (the structures of both bio-surfactants are shown in Fig. 1). We transferred solutions containing commercially available rhamnolipids and surfactin extracts at various concentrations to PDMS chambers with 30° corner angles and visualized the flows inside the chambers (see Materials and Methods for details). The corner flows generated by both biosurfactants at selected concentrations at maximum height are shown (Fig. 1A and B, top right). The surface tension $\gamma$ and contact angle $\theta_c$ of both biosurfactant solutions at various concentrations were measured from the shape of pendant droplets and the angle of moving droplets on the PDMS surface (Fig. 1, insets; see Materials and Methods for details). As shown by the results in Fig. 1, the surface tension $\gamma$ and contact angle $\theta_c$ for rhamnolipids and surfactin solutions decreased with increasing surfactant concentration when the concentrations were in the range of $1 \times 10^{-5}$ to $1 \times 10^{-2}$ M and $1 \times 10^{-5}$ to $1 \times 10^{-4}$ M, respectively. Both rhamnolipids and surfactin changed the surface from hydrophobic to hydrophilic, with $\theta_c$ reduced from about 120° to about 60°. Corner flows were observed for both biosurfactants when the concentration of rhamnolipids was above 1 to $3 \times 10^{-4}$ M and the concentration of surfactin was above 2 to $3 \times 10^{-5}$ M. These observations confirm our hypothesis that the biosurfactant-generated corner flow mechanism is not unique to rhamnolipids and also occurs for surfactin, which has a different structure than rhamnolipids. Thus, biosurfactant-induced corner flows are likely widely present in hydrophobic soils.

**Corner flows at sharp corners generated by *P. fluorescens* and *B. subtilis*.** We investigated whether two distinct soil bacteria, *P. fluorescens* and *B. subtilis*, could generate corner flows by producing biosurfactants to spread in hydrophobic soil. We used three bacterial strains: a rhamnolipids-producing wild-type (WT) *Pseudomonas fluorescens*

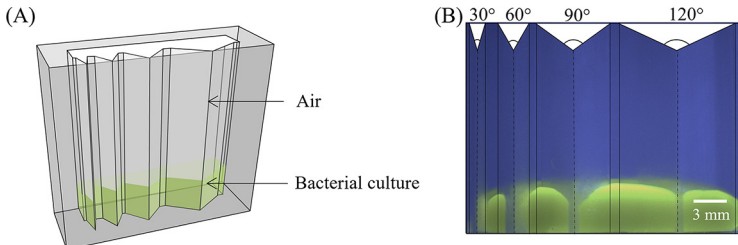

**FIG 2** (A) Schematic of corner flow experiments. Bacterial solution (green) was placed in the PDMS chamber with four corner angles. The green color is due to the addition of 0.005% (wt/vol) fluorescein sodium salt for visualization purposes. The corner angles are 30°, 60°, 90°, and 120° from left to right. (B) Image of the bacterial solution in the chamber at $t = 0$ h, which is defined as the time when the bacterial solution was transferred into the chamber.

strain, a surfactin-producing WT *Bacillus subtilis* strain, and a previously generated surfactin-deficient mutant of *Bacillus subtilis* (43) (see Materials and Methods for details). We grew bacteria in M9 solution in transparent polydimethylsiloxane (PDMS) chambers with four corners with 30°, 60°, 90°, and 120° angles (Fig. 2A). Next, we visualized the spread of the culture medium, which included green fluorescein dye (0.005% concentration, not a surfactant), along these four corners using a digital camera over a 24-h period. Figure 2B shows a representative image of the bacterial solution in the chamber at the time of inoculation ($t = 0$ h). As the bacteria grew in the chamber over time, we observed that both surfactant-producing strains, *P. fluorescens* WT and *B. subtilis* WT, generated corner flows at the 30° corners, as shown in the time-lapse images of the bacterial solution at the 30° corners in Fig. 3A and B. The existence of corner flows at other corner angles and the critical corner angle to generate corner flows will be discussed in the following sections. In contrast to the surfactant-producing strains, the surfactant-deficient mutant of *B. subtilis* did not generate flow at the 30° corners (Fig. 3C). These observations support our hypotheses that surfactant-producing bacteria, such as *P. fluorescens* WT and *B. subtilis* WT, can generate flows along sharp corners by producing surfactants, while a surfactant-deficient mutant of *B. subtilis* cannot.

Second, to investigate the speed of corner flows, we measured the heights of the tips of the corner flows at sharp 30° corners versus time, as shown in Fig. 4. For wild-type *P. fluorescens*, the corner flow started at $t = 8$ h and ended at $t = 15$ h with a maximum height of about 9 mm. For wild-type *B. subtilis*, the corner flow started at $t = 2$ h and ended at $t = 18$ h with a maximum height of 9 mm. Thus, the average climbing speeds were 1.3 mm/h and 0.6 mm/h for wild-type *P. fluorescens* and *B. subtilis*, respectively. The speed of the bacterial corner flow, on the order of millimeters per hour, was similar to bacterial surface swarming, the fastest mode of known bacterial surface translocation (44).

Furthermore, we show that bacterial cells are transported with the bacterially generated corner flows. We sampled the *B. subtilis* solution at the tip of the corner flow using 10-$\mu$L pipette tips after 24 h and diluted it with abiotic M9 solution by about 50 times. Then, we imaged the bacterial sample under a Nikon C2 plus confocal laser scanning microscope. As shown in Fig. S2 in the supplemental material, *B. subtilis* WT cells existed in the tip of the corner flow, suggesting that surfactant-producing bacteria can indeed make use of biosurfactant to spread along corners in hydrophobic soil.

**Contact angle and the critical corner angle for bacterial corner flows.** To demonstrate how bacterially produced biosurfactants generate corner flows, we measured the optical density at 600 nm ($OD_{600}$) and surfactant-related properties, including surface tension $\gamma$ and contact angle $\theta_c$ of the bacterial solution over time. Note that because the volume of the bacterial solution in the PDMS chamber was not sufficient for these measurements, we used bacterial solution grown in 50-mL tubes under identical oxygen, nutrient, and temperature conditions as in the PDMS chamber for measurements. For *P. fluorescens* WT and *B. subtilis* WT, $\theta_c$ and $\gamma$ decreased gradually beginning at $t = 6$ h and $t = 4$ h, respectively (Fig. 5A and B). At the beginning of the experiments ($t = 0$ h), the contact angle of the bacterial solution on the PDMS surface was $\theta_c \approx 115°$ for all strains, because the solution was similar to

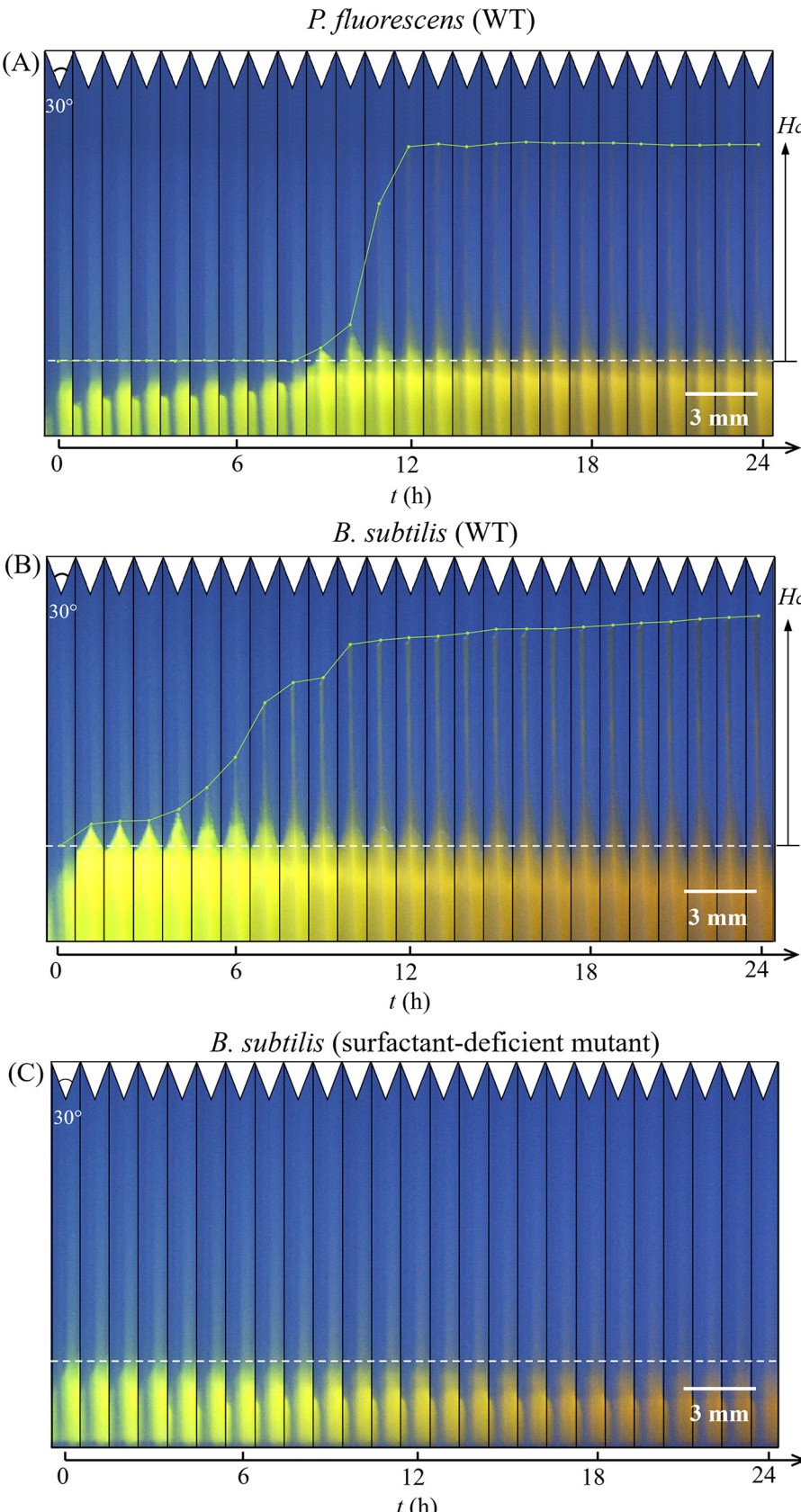

**FIG 3** Time-lapse images of corner flow at the 30° corners during a 24-h bacterial growth period for a WT surfactant-producing strain of *P. fluorescens* (A), a WT surfactant-producing strain of *B. subtilis* (B), and

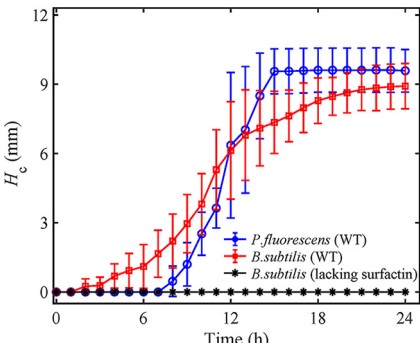

**FIG 4** The evolution over time of the tip positions of corner flows at the 30° corner for *P. fluorescens* WT, *B. subtilis* WT, and a surfactant-deficient mutant of *B. subtilis*. $H_c$ indicates the tip position of the corner flow relative to its initial position (shown in Fig. 3). The error bars represent the standard errors of measurements of six to seven replicate experiments for each strain.

water, whose contact angle on PDMS is about 117° (45). At $t = 16$ h, the wettability of PDMS for both strains, *P. fluorescens* WT and *B. subtilis* WT, changed from initially nonwetting ($\theta_c \approx 115°$) to wetting ($\theta_c \approx 60°$). In comparison, for the surfactant-deficient mutant of *B. subtilis*, despite a slight decrease in surface tension $\gamma$, the contact angle $\theta_c$ of the bacterial solution on the solid surface remained above 90° (Fig. 5C), and thus, the surface remained nonwetting. For the surfactant-producing strain *B. subtilis* WT, $\theta_c$ dropped from 115° (nonwetting) to 60° (wetting) during $t = 4$ to 17 h (Fig. 5B), starting earlier than for *P. fluorescens*, for which $\theta_c$ dropped during $t = 6$ to 17 h (Fig. 5A). Consistently, as shown by the results in Fig. 4, the beginning time of corner flows in chambers for *B. subtilis* WT was around $t = 2$ h, earlier than the start time of the corner flow of *P. fluorescens*, which was around $t = 7$ h. For *P. fluorescens* WT and *B. subtilis* WT, the start times of corner flows were consistent with the time when the contact angle changed from hydrophobic (>90°) to hydrophilic (<90°), further confirming that the bacterial corner flow was induced by the surfactant-induced change in the contact angle of the solution.

Next, we show that the critical corner angle for bacteria to generate corner flow can be predicted from the contact angle of the bacterial solution on PDMS surfaces using classic corner flow theories developed for hydrophilic surfaces and wetting fluids. According to the Concus-Finn criterion (21), corner flow occurs when $(\alpha/2) + \theta_c < (\pi/2)$, where $\alpha$ is the corner angle and $\theta_c$ is the contact angle. For *B. subtilis* WT, after the bacteria produced sufficient surfactants, the advancing contact angle on the PDMS surface reduced to around 60°, as shown by the results in Fig. 5B, so the predicted critical corner angle for corner flow is $\alpha_{th} = 2 \times (\pi/2 - \theta_{min}) = 60°$, where $\alpha$th is the threshold angle of corner to generate corner flow and subscript th is the abbreviation of threshold. We prepared PDMS chambers that contained interior corners of different degrees, including 30°, 50°, 60°, 90°, and 120°. Afterward, we transferred $700 \pm 100$ $\mu$L bacterial culture at an $OD_{600}$ of $0.5 \pm 0.1$ into the chamber and imaged the position of the bacterial solution over 24 h. During the experiments, we observed noticeable corner flows along corners with 30° and 50° corners, but no

**FIG 3** Legend (Continued)

a surfactant-deficient mutant of *B. subtilis* (C) (see Materials and Methods for details). Images were cropped at the 30° corner from time sequence images of the chamber with 4 different angles (30°, 60°, 90°, and 120°) shown in Fig. 2B. The white dashed horizontal lines represent the initial height of bacterial culture in the chamber at growth time $t = 0$ h. $H_c$ represents the height of corner flows at different times relative to the original location at $t = 0$ h. The green lines were added to the images to indicate the tip positions of the corner flows over time. The green color of the bacterial solution is from the 0.005% fluorescein sodium salt added to the bacterial culture. The color of bacterial culture in the chamber gradually turned from bright green to dark yellow due to the increase in bacterial cell density, which made the solution turbid; examples of the color change are shown in Fig. S1. Note that the contrast and brightness of the figures have been enhanced to increase the visibility of corner flow. Videos of corner flow development with the original colors are shown in Movie S1, Movie S2, and Movie S3.

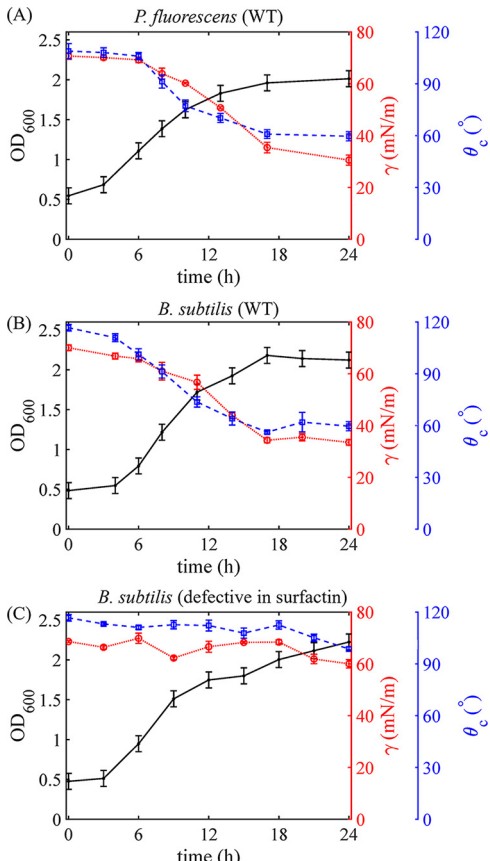

**FIG 5** The evolution over time of the advancing dynamic contact angle $\theta_c$ on a PDMS surface (blue, right *y* axis), the surface tension $\gamma$ (red, right *y* axis), and the $OD_{600}$ (black, left *y* axis) of bacterial solutions of *P. fluorescens* WT, *B. subtilis* WT, and a surfactin-deficient mutant of *B. subtilis*. The error bars represent the standard errors of measurements of 3 to 4 liquid drops.

fluid rises along 90° or 120° corners (Fig. 6). Note that the triangle-shaped liquid rise at the 90° or 120° corners was not corner flow but instead was the adjustment of the air-water interface due to the change in contact angle. The corner flow is the rise of liquid above the triangle-shaped liquid. A slight rise above the triangle-shaped air-water interface at the 60° corner was also observed, indicating that the critical corner angle for *B. subtilis* WT was about 60°. This observed critical corner angle is consistent with the predicted critical corner angle $\alpha_{th} = 60°$ calculated by classic corner flow theory. For *P. fluorescens* WT, the contact angle was also reduced to 60°, and thus, the predicted critical corner angle was also $\alpha_{th} = 60°$. Our experiments in the chamber with four different angles (30°, 60°, 90°, and 120°) (Movies S1 to S3 are experiments in chambers with 4 corners, Movie S4 is conducted in chambers with 3 corner angles in the supplemental material) also showed corner flows at the 30° corner but no fluid rise at the 90° or 120° corner. The agreement of the predicted and the observed critical corner

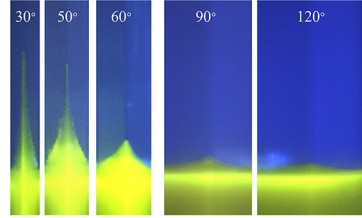

**FIG 6** Images of solutions containing *B. subtilis* WT cells in five different corners after a 20-h growth period. The initial bacterial density was an $OD_{600}$ of ≈0.5 ± 0.1. The first three images, with corners of 30°, 50°, and 60°, were cropped from imaging of one chamber (as shown in Movie S4), and the last two images, with corners of 90° and 120°, were cropped from imaging of the chamber shown in Fig. 2A.

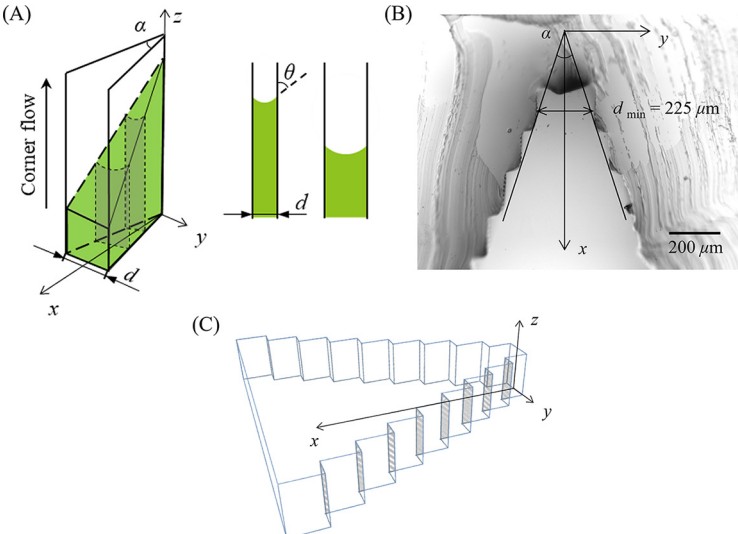

**FIG 7** (A) A schematic diagram of the upward flow of the bacterial solution (green) along a 30° corner. A cross-sectional image parallel to the (y, z) plane is shown at the right. $\theta_c$ is the contact angle, $\alpha$ is the corner angle, and $d$ is the local separation changing with the distance $x$. (B) Confocal image of a cross-section of the 30° corner of the PDMS chamber. The minimum separation in the tip position is 225 $\mu$m. (C) A schematic of the step-like structures of the 30° corner.

angle for both surfactant-producing bacteria, *P. fluorescens* WT and *B. subtilis* WT, further confirms that the bacterial corner flow is due to and can be predicted by biosurfactant-induced changes in the contact angle.

**Maximum height of corner flow.** During our experiments, we observed that the tip positions of the corner flows generated by *P. fluorescens* WT and *B. subtilis* WT at the 30° corner stopped at a maximum height, $h_{max} \approx 9$ mm, as shown by the results in Fig. 4. However, corner flows can theoretically rise to infinite height if the corner is perfectly sharp (46, 47). We hypothesize that the maximum corner flow height is limited by the roundness or cutoff of the corner, because corners can hardly be perfectly sharp (48, 49). The microscopic image of the 30° corner (Fig. 7B) shows that the corner in our experiments, due to the resolution of our three-dimensional (3-D) printing (see Materials and Methods for details), was rounded and had step-like structures along its inner surface, as shown in Fig. 7C. Understanding how corner roundness controls the height of bacterial corner flows (48, 50) is important, because natural corners are often not perfectly sharp.

To estimate the maximum height of corner flow along the rounded corner, we used a classic fluid mechanics theory developed for pure wetting liquid (18, 51). The following assumptions were made (52, 53): (i) the motion of the fluid is mainly vertical and dominated by the curvature in the (y, z) plane and the constant capillary pressure can be calculated from the contact angle $\theta_c$, and (ii) evaporation of liquid, friction, and inertial effects are not considered. Imagine that the two plates that form the corner are composed of many parallel plates with different distances $d$ apart, e.g., they consist of steplike structures in the inner surface of the chamber (Fig. 7C). The fluid meets the container wall with a prescribed contact angle $\theta_c$. The weight of liquid that rises vertical distance $h$ by a segment of the wall of length $l$ between two parallel walls with distance $d$ is $F_g = \rho gdhl$, where $g$ is the gravitational acceleration and $\rho$ is the liquid density. The capillary force due to surface tension of the liquid between two parallel plates can be derived using the Young-Laplace equation (54), $F_\sigma = 2l\gamma\cos\theta_c$. Due to force balance, the capillary driving force ($F_\sigma$) is equal to the gravitational force ($F_g$), and thus, the maximum height of the liquid rise is as follows:

$$h = \frac{2\gamma\cos\theta_c}{\rho gd}$$

From the microscopic image of the corner (Fig. 7B), the minimum width at the tip of the corner is $d = 225$ $\mu$m, which is the minimum width of a series of parallel plates. Substituting

$\theta_c = 60°$, $\gamma = 30$ mN/m, $\rho = 1{,}000$ kg/m³, and $d = 225$ $\mu$m into the equation above, we found that $h = 13.6$ mm. Consistently, our experimental results showed that the maximum height of the corner flow at the 30° corner was 9 mm for the two surfactant-producing bacterial species considered here, which is the same order of magnitude as our theoretical estimation, suggesting that the maximum height of corner flow is indeed limited by the roundness of the corner.

## DISCUSSION

We demonstrate that two common types of biosurfactants, rhamnolipids and surfactin, can generate flows along sharp corners where two initially hydrophobic surfaces meet by changing the surface into a hydrophilic one. Furthermore, we show that *Pseudomonas fluorescens* and *Bacillus subtilis*, soil bacteria and producers of rhamnolipids and surfactin, respectively, can generate corner flows, while a surfactant-deficient mutant cannot. The corner flow only occurs when the corner angle is less than a critical value, which we provide models to predict based on classic fluid mechanics theories. Furthermore, we show that the maximum height of the corner flow is limited by the roundness of the corner geometry and can also be predicted from fluid mechanics theories. The biosurfactant-induced corner flows that we documented, using two common biosurfactants that have distinct chemical structures and two different types of soil bacteria (one Gram negative and the other Gram positive), suggest that biosurfactant-induced corner flows and associated bacterial spreading are likely widely present in hydrophobic soil, especially hydrocarbon-contaminated soil near oil fields (29, 55–59). The mechanistic understanding and mathematical characterization of bacterial corner flows that we have developed will help improve predictions of bacterial spreading in hydrophobic soil (57, 58) and facilitate the design of soil-based bioremediation projects (60–62), such as the use of bacteria to remediate oil-contaminated soil (63, 64).

## MATERIALS AND METHODS

**Biosurfactant information.** The two biosurfactants, rhamnolipids and surfactin, used in this study were purchased from Sigma-Aldrich. The rhamnolipids (catalog number R90; Sigma-Aldrich) were purified from *Pseudomonas aeruginosa* and contain a 90% mixture of rhamnolipid congeners. The surfactin (catalog number S3523; Sigma-Aldrich) was 98% pure and extracted from *Bacillus subtilis*.

**Bacterial strains and growth.** The bacterial strains used in the present study were *Pseudomonas fluorescens* PF15 (wild type), *Bacillus subtilis* 3610 (wild type), and *Bacillus subtilis* DS1122 (defective in surfactin production). *Pseudomonas fluorescens* PF15 was donated by Mohamed Donia. *B. subtilis* DS1122 was a mutant (*srfAC*::*Tn*10 *spec*) of wild-type *B. subtilis* 3610 with a transposon inserted into a gene required for surfactin production (43). Bacterial cells were streaked from −80°C freezer stocks onto Luria-Bertani (LB) plates (1.5% agar). *P. fluorescens* and *B. subtilis* were grown at 30°C and 37°C, respectively.

**Bacterial solution.** The initial bacterial solution, at an $OD_{600}$ of 0.5 $\pm$ 0.1 in M9 solution, used for the corner flow experiments, was prepared following the steps described below. First, 5 mL of LB liquid medium in a 50-mL tube was inoculated with cells from an isolated colony on the plate. The LB medium inoculated with *P. fluorescens* was placed on a shaker at 30 $\pm$ 2°C and 200 rpm and the LB medium inoculated with *B. subtilis* was incubated on a shaker at 37 $\pm$ 2°C and 200 rpm overnight for 16 h to reach the exponential-growth phase (65, 66). Second, the bacterial overnight cultures were subjected to centrifugation at 4,000 rpm for 10 min. Afterward, we removed the supernatant, diluted the bacterial cells at the bottom of the tube with M9 medium, and mixed them using a vortex mixer. This step was repeated twice to remove any residual surfactant on cell surfaces. The cell density of the culture was diluted to an $OD_{600}$ of 0.5 $\pm$ 0.1 by adjusting the volume of M9 medium. The M9 medium used in this study was supplemented with 0.03 $\mu$M $(NH_4)_6(Mo_7)_{24} \cdot 4H_2O$, 4 $\mu$M $H_3BO_3$, 0.3 $\mu$M $CoCl_2 \cdot 6H_2O$, 0.1 $\mu$M $CuSO_4 \cdot 5H_2O$, 0.8 $\mu$M $MnCl_2 \cdot 4H_2O$, 0.1 $\mu$M $ZnSO_4 \cdot 7H_2O$, 0.1 $\mu$M $FeSO_4 \cdot 7H_2O$, and 2% glucose. When noted, 0.005% (wt/vol) fluorescein sodium salt was added.

**Preparation of 3-D-printed molds and fabrication of the PDMS slabs.** We used 3-D-printed molds to produce the PDMS slabs used in the experiment. The molds were composed of a cuboid (30 mm by 25 mm by 4 mm) and four triangular prisms (an image of the mold is shown in the supplemental material). The heights of the cross sections of these triangular prisms were all 3 mm. The mold with the four different angles was printed by a 3-D printer (Anycubic Photon Mono X) using a 405-nm UV resin (Anycubic). The printed molds could not be immediately used for PDMS casting because chemicals released from 3-D-printed objects will inhibit PDMS curing in the vicinity of these objects (67). To avoid this problem, the printed mold was UV cured for 20 min, immersed in 99% isopropanol for 6 h, treated with air plasma corona (BD-20AC laboratory corona treater) for 1 min, and then silanized using triethoxy(1*H*,1*H*,2*H*,2*H*-perfluoro-1-octyl)silane for 3 h. Then, the mold was transferred to a petri dish and a 10:1 (wt/wt) base/curing agent PDMS liquid was poured onto the 3-D printed mold. The composite was cured in a hot plate at 80°C for at least 2 h, and the PDMS chamber was soaked in 70% isopropanol for about 18 h to keep it sterilized.

**Corner flow experiment.** For the corner flow experiment, a sterilized PDMS chamber was placed in an incubator with a transparent front door (Fig. 8). We transferred 700 $\pm$ 100 $\mu$L prepared bacterial culture at an

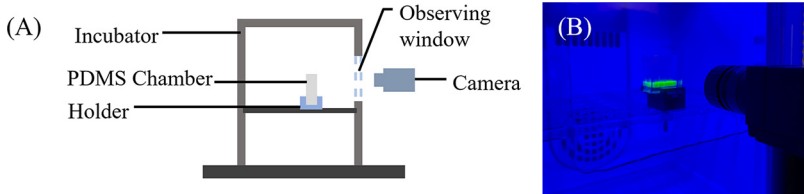

**FIG 8** (A) A schematic diagram of the bacterial corner flow experimental setup. (B) An image of the experimental setup for the bacterial corner flow experiments.

$OD_{600}$ of $0.5 \pm 0.1$ into the PDMS chamber using a 3-mL syringe. The incubator was set to a temperature of $30 \pm 2°C$ for *P. fluorescens* and $37 \pm 2°C$ for *B. subtilis*, and the relative humidity was kept at $80\% \pm 10\%$. To visualize the bacterially induced corner flow, a blue LED light was placed in the incubator and the M9 medium contained fluorescent 2-NBDG {2-[*N*-(7-nitrobenz-2-oxa-1,3-diazol-4-yl)amino]-2-deoxyglucose} to make the slender corner flow visible. A digital camera (Blackfly S BFS USB3; Teledyne FLIR) was placed in front of the chamber and set to take photos at 2-min intervals for 24 h. The height of the corner flow was measured from the initial water surface level to the top of the corner flow.

**Contact angle and surface tension measurements.** To quantify the biosurfactant-related parameters, we measured the time evolution of contact angle $\theta_c$, surface tension $\gamma$, and bacterial cell density ($OD_{600}$) over time for 24 h. Due to the limited volume of bacterial solution in the PDMS chamber, we grew bacterial solutions in 50-mL tubes to mimic the growth of bacteria in chambers. We transferred the prepared uniform bacterial cultures ($OD_{600} = 0.5 \pm 0.1$) by 5-mL aliquots into multiple 50-mL centrifuge tubes. We placed the tubes containing 5-mL cultures in a shaking incubator. Specifically, the temperature was set to $30 \pm 2°C$ for *P. fluorescens* and $37 \pm 2°C$ for *B. subtilis*. For each data point in Fig. 5, we removed one tube from the incubator, transferred 1 mL of bacterial culture to a cuvette, and measured the $OD_{600}$. The initial $OD_{600}$ was diluted to $0.5 \pm 0.1$, so the error bar/uncertainty of each $OD_{600}$ data point (Fig. 5) was set to 0.1.

To measure surface tension $\gamma$ and contact angle $\theta_c$, bacterial culture was centrifuged at 4,000 rpm for 10 min and the supernatant was filtered through a 0.2-$\mu$m filter to remove bacterial cells. Afterward, we transferred the filtered solution into a 10-mL syringe with an 18-gauge needle. We placed the syringe on a syringe pump connected to a needle, and a drop of the solution was pushed out of the needle. To measure surface tension $\gamma$, the shape of the pendant drop below the needle was recorded, and the profile of the droplet edge was fitted using the MATLAB code developed by the Stone group based on the algorithm proposed by Rotenberg et al. (68). To measure the advancing contact angle $\theta_c$, bacterial solution was pushed out of the needle onto the surface of a PDMS slab and the shape of the advancing drop imaged, using the syringe pump at a 1.2-mL/h flow rate. After identifying the edges of the moving drops, we estimated $\theta_c$ as the angle between the PDMS surface and the tangent line of the drop edge near the contact line. Examples of pendant drop and advancing drop are shown in Fig. 1.

**Data availability.** The MATLAB codes for image processing and the estimation of surface tension and contact angle have already been shared by J. Q. Yang on GitHub (https://github.com/JudyQYang/Bacterial _corner_flow_codes). All other study data were extracted from Movies S1 to S4 in the supplemental material and are available from the corresponding author upon reasonable request.

## SUPPLEMENTAL MATERIAL

Supplemental material is available online only.
**SUPPLEMENTAL FILE 1**, AVI file, 18.8 MB.
**SUPPLEMENTAL FILE 2**, AVI file, 17.9 MB.
**SUPPLEMENTAL FILE 3**, AVI file, 17.1 MB.
**SUPPLEMENTAL FILE 4**, AVI file, 17.4 MB.
**SUPPLEMENTAL FILE 5**, PDF file, 0.1 MB.

## ACKNOWLEDGMENTS

This research was supported by J. Q. Yang's startup fund and the Minnesota Environment and Natural Resources Trust Fund as recommended by the Legislative-Citizen Commission on Minnesota Resources (LCCMR). Y. Li was supported by the fellowship of Civil, Environmental, and Geo-Engineering at the University of Minnesota. J. Sanfilippo was supported by NIH grant no. K22 AI151263. D. Kearns was supported by NIH grant no. R35 GM131783. We thank Mohamed Donia (Princeton University) for sharing with us the bacterial strain *Pseudomonas fluorescens* PF15.

J. Q. Yang and Y. Li conceived the idea, designed the research, and wrote the paper. Y. Li designed and conducted the experiments under the guidance of J. Q. Yang. J. Sanfilippo and D. Kearns contributed strains and contributed to the research idea and writing.

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
