## [Reviewer comments · Microbiology Spectrum]

Microbiology Spectrum

Corner flows induced by surfactant-producing bacteria *Bacillus subtilis* and *Pseudomonas fluorescens*

Yuan Li, Joe Sanfilippo, Daniel Kearns, and Judy yang

Corresponding Author(s): Judy yang, University of Minnesota-Twin Cities

Review Timeline:

Submission Date:	September 12, 2022
Editorial Decision:	September 20, 2022
Revision Received:	September 20, 2022
Accepted:	September 23, 2022

Editor: Blaire Steven

Reviewer(s): The reviewers have opted to remain anonymous.

Transaction Report:

DOI: <https://doi.org/10.1128/spectrum.03233-22>

September 20, 2022

Dr. Judy yang
University of Minnesota-Twin Cities
2 SE 3rd Ave
Minneapolis

Re: Spectrum03233-22 (Corner flows induced by surfactant-producing bacteria *Bacillus subtilis* and *Pseudomonas fluorescens*)

Dear Dr. Judy yang:

Link Not Available

Sincerely,

Blaire Steven

Journals Department
Reviewer comments:

Staff Comments:

Preparing Revision Guidelines

- Point-by-point responses to the issues raised by the reviewers in a file named "Response to Reviewers," NOT IN YOUR

COVER LETTER.

- Upload a compare copy of the manuscript (without figures) as a "Marked-Up Manuscript" file.
- Each figure must be uploaded as a separate file, and any multipanel figures must be assembled into one file.
- Manuscript: A .DOC version of the revised manuscript
- Figures: Editable, high-resolution, individual figure files are required at revision, TIFF or EPS files are preferred

Please return the manuscript within 60 days; if you cannot complete the modification within this time period, please contact me. If you do not wish to modify the manuscript and prefer to submit it to another journal, please notify me of your decision immediately so that the manuscript may be formally withdrawn from consideration by Microbiology Spectrum.

September 23, 2022

Dr. Judy yang
University of Minnesota-Twin Cities
2 SE 3rd Ave
Minneapolis

Re: Spectrum03233-22R1 (Corner flows induced by surfactant-producing bacteria *Bacillus subtilis* and *Pseudomonas fluorescens*)

Dear Dr. Judy yang:

Your manuscript has been accepted, and I am forwarding it to the ASM Journals Department for publication. You will be notified when your proofs are ready to be viewed.

Sincerely,

Blaire Steven
Editor, Microbiology Spectrum

Journals Department
Supplemental Movie S4: Accept
Supplemental Movie S2: Accept
Supplemental Movie S1: Accept
Supplemental Movie S3: Accept
Supplemental Material: Accept